# Racial disparities in urine drug screening among seizure patients in the emergency department

**Gloria Ortiz-Guerrero**[ID]¤*, **Maggie Logan, Utku Uysal, Patrick Landazuri**[ID],
**Lakeshia Jackson**[ID]**, Carol M. Ulloa**

Department of Neurology, The University of Kansas Medical Center, Kansas City, Kansas, United States of America

¤ Current address: Department of Neurology, Epilepsy Division, Mayo Clinic, Rochester, MN, USA
* ortizguerrero.gloria@mayo.edu

## Abstract

### Background

The potential for racial disparity using urine drug screening (UDS) in patients with seizures is sparsely reported. This study aims to determine racial and ethnic disparities when ordering UDS in patients with suspected seizures in the emergency department (ED).

### Methods

In this retrospective study, we identified patients over the age of 18 with suspected seizures who presented to the ED at the University of Kansas Medical Center between October 2017 and October 2020. Data encompassed demographic, clinical, electrographic, and UDS information from the electronic medical records (EMR). Data were compared between Black and White patients, as well as between Hispanic and non-Hispanic patients. We used the Chi-square test to assess differences in UDS testing based on patient race and ethnicity.

### Results

A total of 2945 patients were identified. Among these, 1750 (59%) were White and 821 (28%) were Black patients. Of these patients, 612 (20.8%) underwent UDS in the ED. Black patients had UDS performed more frequently compared to White patients ($p=0.046$). Additionally, the positivity rate for UDS was higher among Black patients than White patients ($p=0.019$). UDS testing did not differ by ethnicity ($p=0.164$). Amphetamine positivity was higher among White patients; however, the positivity rates of other drugs did not differ by race and ethnicity. Black patients were more often admitted to the neurology intensive care unit. No differences in discharge rates or EEG testing were noted across race or ethnicity.

**Data availability statement:** All relevant data are within the manuscript.

**Funding:** P. Landazuri is part of the Medical Advisory Board and consultant for NeuroPace. He receives royalties from Cambridge University Press for a book. U. Uysal serves as a site principal investigator for a multicenter trial sponsored by NeuroPace and institutional funding was provided to support study conduct. C. Ulloa has received research funding from Medtronic (all funds to University of Kansas Medical Center). The remaining authors have no disclosures.

**Competing interests:** The authors have declared that no competing interests exist.

## Conclusion

Increased UDS testing was seen in Black patients compared to White patients in our single-institution cohort, warranting further research into the underlying causes of this disparity.

## Introduction

In 2021 and 2022, an estimated 2.9 million U.S. adults and 456,000 children under 18 had active epilepsy [1,2], accounting for approximately 1% of the adult population [1]. Seizures account for approximately 1–2% of Emergency Department (ED) visits in the U.S. [3]. Racial and ethnic minorities, especially Black patients, are disproportionally affected by higher seizure burden [4], increased rates of hospitalization and ED visits, and reduced adherence to antiseizure medications (ASMs) compared to White patients and individuals with higher socioeconomic status (SES) [4,5]. Moreover, Black patients are more likely to have non-private insurance and less access to outpatient care, contributing to more frequent ED visits within a calendar year [3,6]. These repeat ED visits are primarily associated with missing or running out of ASMs [6]. In a pediatric study, Black children who were admitted to the hospital had 42% higher odds of drug testing compared to White children, although ED-only testing rates were similar [7]. Urine drug screening (UDS) in patients presenting to the ED with seizures is at the treating physician's discretion. Although literature evaluating racial disparities in seizure care is rising, data addressing racial disparities in ED drug testing in seizure patients is still sparse. This study aims to assess UDS rates by race and ethnicity in patients presenting to the ED with seizures in a single academic medical center. We also analyze UDS positivity rates, drug use prevalence, hospital admission rates, and EEG utilization across racial and ethnic groups.

## Methods

The local institutional review board (IRB) approved this study on October 20, 2020 (IRB STUDY00146470). Since our group utilized deidentified data, this study was deemed exempt, and the IRB waived the requirement for informed consent. The data was accessed for research purposes from October 20, 2020, to June 30, 2021. This retrospective study included patients aged 18 years and older, who visited the ED at the University of Kansas Medical Center (KUMC) between October 1, 2017, and October 31, 2020, for seizures and underwent UDS. Seizure-related diagnosis were identified using the International Classification of Diseases (ICD), 10th (ICD-10) codes. Once these patients were identified, we extracted demographic, clinical, electrographic, and UDS data from the electronic medical records (EMR). If UDS data were incomplete or absent, patients were excluded.

Our HERON database [8] automatically categorized demographic information by age, gender, language, race, and ethnicity. Races were classified as White, Black, American Indian, Asian, Pacific Islander, Other, Declined, Two Races, or Undetermined. Ethnicity included Hispanic, non-Hispanic, Declined, or Undetermined.

This article uses the *AMA Manual of Style* terminology to refer to race and ethnicity: White, Black, Hispanic, and non-Hispanic [9].

Data were collected from the five most recent ED visits. Urine toxicology results were stratified by age, gender, race, ethnicity, and language. UDS included amphetamines, barbiturates, benzodiazepines, cannabinoids, cocaine, opiates, and phencyclidine. The UDS was positive if the drug was detected in the urine sample regardless of the level. UDS testing and positivity rates by race and ethnicity were all recorded.

EEG testing was categorized based on duration into three categories: 25-minute EEG, one-hour EEG, and 24-hour video EEG. The EEG categories according to the findings were normal, generalized slowing, focal slowing, periodic pattern, seizures, status epilepticus, and non-epileptic events. The final patient's disposition was categorized as hospital admission, discharge, or leave against medical advice. Hospital admissions were further divided by admitting service: internal medicine, neurology, psychiatry, trauma, family medicine, medical intensive care unit (ICU), neurology ICU, neuro-surgery, cardiology, transplant surgery, and oncology.

## Statistical analysis

Demographic characteristics were described with frequency measures, means, and standard deviations. Kruskal Wallis test was used to compare ages among Black, White, and "Other" races. The Mann-Whitney test compared the ages of Hispanic and non-Hispanic patients. Differences in UDS regarding positivity rates and repeated UDS frequency were evaluated using the Chi-square test for two categorical variables. A p-value of ≤ 0.05 was considered statistically significant. To enhance clinical interpretability, 95% confidence intervals were calculated for proportions, and effect sizes were considered where relevant. IBM SPSS Statistics (Version 27) was used for data analysis.

## Results

From October 1, 2017, to October 31, 2020, 2945 patients older than 18 were evaluated in the ED at KUMC for seizures regardless of etiology. Among these patients, 1750 (59%) were White and 821 (28%) were Black; 291 (10%) considered themselves as "Other" race. The remaining races (3%) were Asian, Native American, and Pacific Islanders. Patients who noted declined, undetermined, and two races were included in remaining races. Regarding ethnicity, 2652 (90%) were non-Hispanics and 257 (9%) were Hispanics. Of 2945 patients, 612 (20.8%) had UDS. The demographic characteristics of these groups are shown in Table 1.

UDS testing and positivity rates are shown in Figs 1 and 2. Black patients underwent UDS more than White patients (23% vs. 20%, $p = 0.046$) and were more likely to have positive results (59% vs. 48%, $p = 0.019$). No differences in toxicology testing rates were observed between Hispanic and non-Hispanic patients (62/257–24% vs. 542/2652–20%, $p = 0.164$). However, Hispanic patients were less likely to have a positive result than non-Hispanic patients (34% vs. 52%, p = 0.007).

**Table 1. Demographic data of patients evaluated in the ED for seizures compared to the subgroup of UDS patients.**

| Characteristic | All Patients (n = 2945) (95% CI) | UDS Tested (n = 612) (95% CI) |
|---|---|---|
| Age yr(mean ± SD) | 49.24 ± 18.64 [48.57, 49.91] | 47.51 ± 18.30 [46.06, 48.96] |
| Gender- females | 1409 (47.8%) [0.46, 0.50] | 262 (42.8%) [0.39, 0.47] |
| English speakers | 2818 (95.7%) [0.95, 0.96] | 572 (93.5%) [0.91, 0.95] |
| Spanish speakers | 84 (2.9%) [0.02, 0.04] | 26 (4.2%) [0.03, 0.06] |
| White | 1750 (59.4%) [0.58, 0.61] | 343 (56.0%) [0.52, 0.60] |
| Black | 821 (27.9%) [0.26, 0.30] | 189 (30.9%) [0.27, 0.35] |
| Non-Hispanics | 2652 (90.1%) [0.89, 0.91] | 542 (88.6%) [0.86, 0.91] |
| Hispanics | 257 (8.7%) [0.08, 0.10] | 62 (10.1%) [0.08, 0.13] |

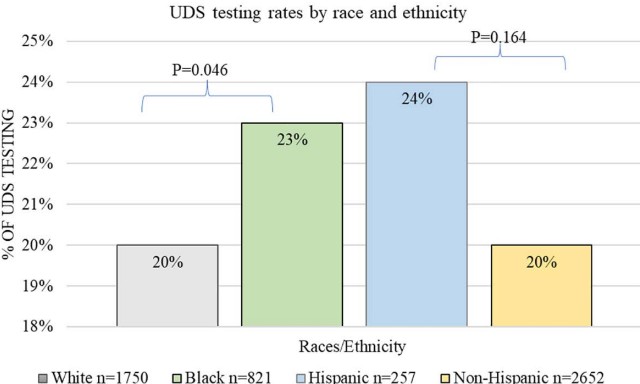

**Fig 1. UDS testing rates by race and ethnicity.** Black patients most frequently underwent UDS in the ED compared to White patients. No differences were observed by ethnicity.

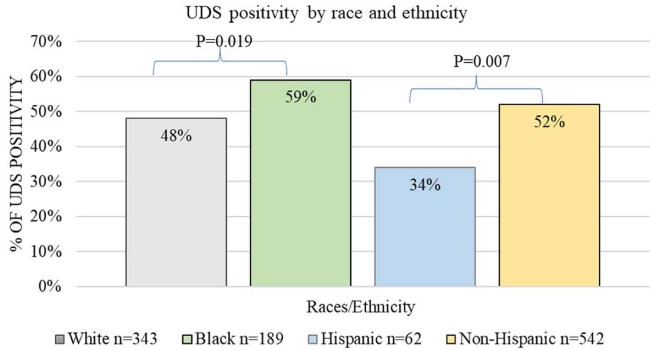

**Fig 2. UDS positivity rates by race and ethnicity.** UDS positivity rates were highest among Black patients, and, by ethnicity, among non-Hispanic individuals.

Amphetamine positivity was higher in White patients; however, no racial and ethnic differences were appreciated for other substances. Additional details are shown in Table 2.

No significant differences in frequency were observed during the first three ED visits by race. However, by the fourth visit, frequency was significantly higher in Black patients ($p = 0.00001$). Details about ED visit rates are shown in Fig 3. Regarding language, UDS positivity rates were higher in English-speaking patients than in Spanish speakers (51% vs.

**Table 2. Distribution of positive drug types by race and ethnicity.**

| Substance | White 164/343 | White 95% CI | Black 111/189 | Black 95% CI | p-value |
|---|---|---|---|---|---|
| Benzodiazepines | 60 (36.6%) | [0.30, 0.44] | 39 (35.1%) | [0.27, 0.44] | 0.9062 |
| Marijuana | 44 (26.8%) | [0.21, 0.34] | 42 (37.8%) | [0.29, 0.47] | 0.0720 |
| Amphetamine | 28 (17.1%) | [0.12, 0.24] | 7 (6.3%) | [0.03, 0.12] | 0.0145 |
| Barbiturates | 16 (9.8%) | [0.06, 0.15] | 7 (6.3%) | [0.03, 0.12] | 0.4284 |
| Cocaine | 8 (4.9%) | [0.02, 0.09] | 11 (9.9%) | [0.06, 0.17] | 0.1701 |
| Opiates | 6 (3.7%) | [0.02, 0.08] | 1 (0.9%) | [0.00, 0.05] | 0.2475 |
| Phencyclidine | 2 (1.2%) | [0.00, 0.04] | 4 (3.6%) | [0.01, 0.09] | 0.2251 |

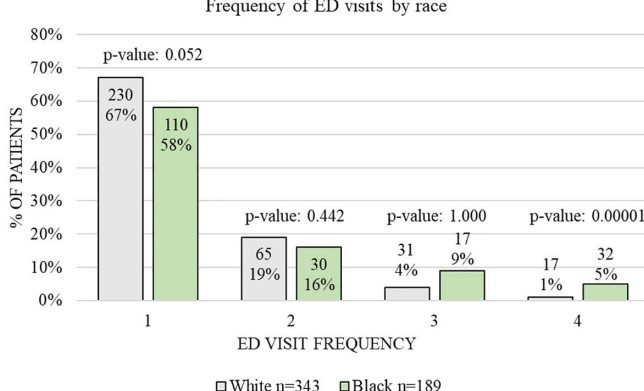

**Fig 3. Frequency of ED visits by race.** The frequency of ED visits became significantly different by the fourth visit, with Black patients presenting most frequently to the ED for seizure evaluation.

27%, $p = 0.015$). In addition, English speakers were more likely to have UDS in the ED than Spanish speakers (93% vs. 4%, $p = 0.017$). Among English and Spanish speakers, there was a tendency for higher positive rates for benzodiazepines (35% vs. 43%, $p = 0.177$). In English speakers, marijuana was the second most frequent drug (32%), while for Spanish speakers, it was barbiturates (29%).

## Post ED course

No difference in disposition was observed between White and Black patients after ED evaluation (69% vs. 65%, $p = 0.253$) or ED discharge rates (31% vs. 35%, $p = 0.289$). Black patients were most likely to be admitted to the neurology intensive care unit (NICU) than White patients (18% vs. 10%, $p = 0.019$). Overall, patients were most frequently admitted to internal medicine ($p = 0.275$), followed by the neurology service ($p = 0.766$); however, no differences were observed across racial groups. Ethnicity did not affect admissions (68% vs. 79%, p = 0.0814) or ED discharges (32% vs. 21%, $p = 0.086$). Hispanic patients were most frequently admitted to the internal medicine (38%) or neurology (21%) services. Only one Black patient was discharged against medical advice from the ED.

No differences in EEG testing rates were observed by race ($p = 0.947$) or ethnicity ($p = 0.585$). Abnormal EEGs were obtained in 78 White patients (67%) and 40 Black patients (62%), whereas by ethnicity, it was abnormal in 118 (64%) non-Hispanics and 13 (68%) Hispanic patients. See Fig. 4 for further details of the EEG testing by race and ethnicity.

## Discussion

Understanding the racial and ethnic composition of the population served is crucial for interpreting the findings of our study. Kansas City, situated between the states of Missouri (MO) and Kansas (KS), has a diverse racial and ethnic distribution. According to recent demographic data, the predominant racial groups in Kansas City, MO, are White (non-Hispanic) ~57%, Black or African American (non-Hispanic) ~26%, Hispanic or Latino—~12%, two or more races (non-Hispanic)—~9%, and Asian (non-Hispanic)—~3% [10]. The University of Kansas Medical Center also serves patients from surrounding states, which may contribute to variations in racial representation in our dataset.

Within this demographic context, our study found that Black patients were more likely to undergo UDS in the ED when presenting with seizures as compared to White patients; however, no difference was noted for ethnicity. Positive UDS results were higher in Black patients and non-Hispanic patients with variability between races for the most common substance positivity testing. Black patients visited the ED more frequently and were most often admitted to the NICU. Several studies have found similar racial and ethnic disparities in adult and pediatric populations evaluated for conditions

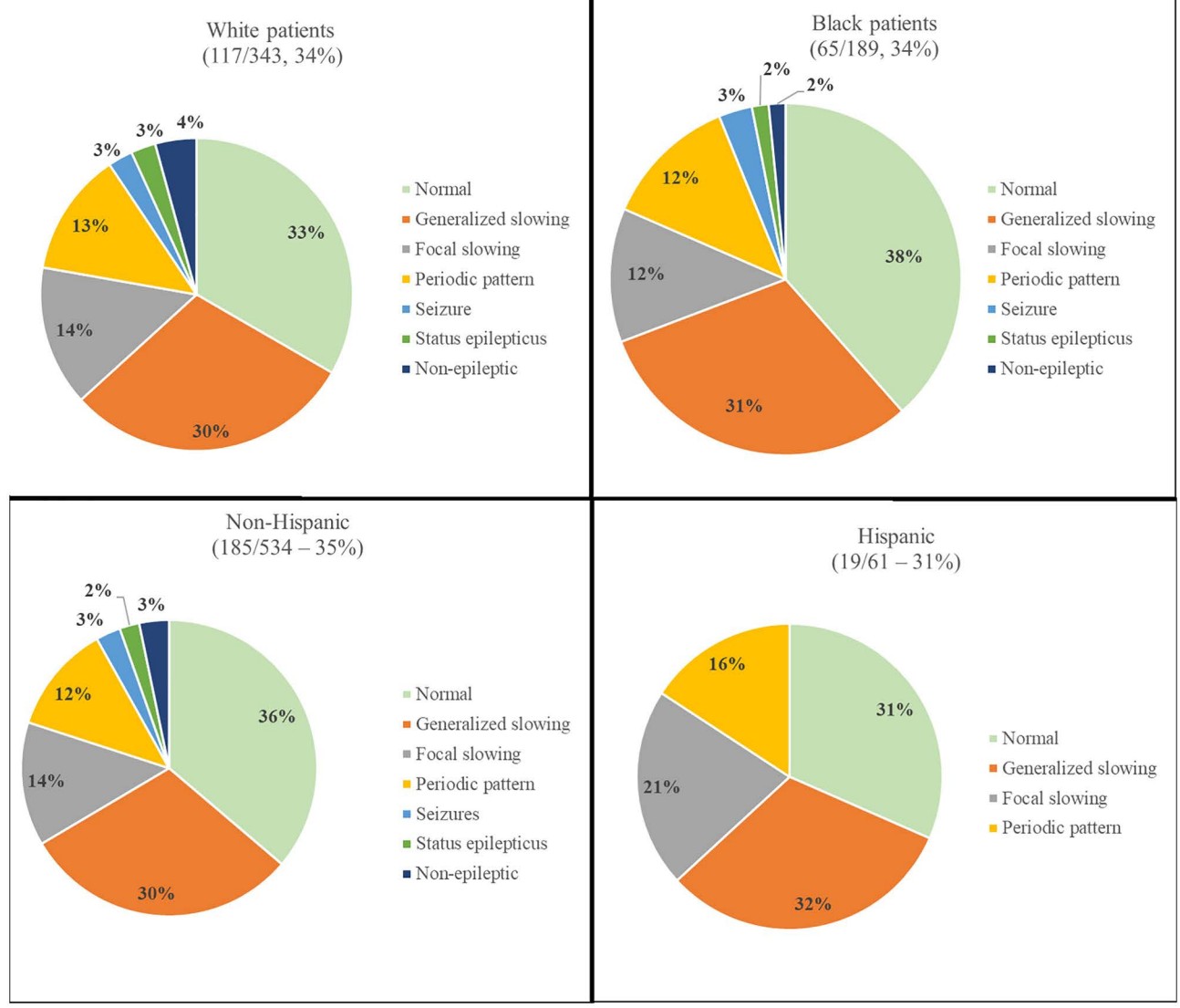

**Fig 4. EEG results divided by race and ethnicity.** No differences in EEG utilization were found among Black, White, non-Hispanic, and Hispanic patients.

unrelated to seizures, particularly when undergoing urine or serum toxicology screen [7,11–14]. In our cohort, we could not identify causality for the higher testing rates in Black patients compared to White patients. However, positivity rates were also higher in Black patients possibly leading to more UDS in this population. Implicit biases among healthcare providers may contribute to the perception that Black and Hispanic patients are at risk for higher substance use, regardless of clinical presentation or current condition. This may lead to higher rates of ordering UDS for these minority groups, perpetuating a cycle and reinforcing stereotypes. In addition, protocols for UDS testing can vary across hospitals, which is at the ordering physician's discretion.

Our study found a higher rate of positive UDS results in Black patients compared to White patients. Although there was a trend toward higher marijuana positivity in Black patients and higher benzodiazepine positivity in White patients, no differences by drug type were observed across racial and ethnic groups —except for amphetamines, which were

significantly most prevalent in White patients. By the time this study concluded, recreational drugs were still illegal in KS and MO, but medical marijuana became legal in MO in 2018 with subsequent recreational legality in 2022, notably with easy access to these sales regardless of which side of the MO/KS state line a patient lives [15]. The legality of medical marijuana in our region sheds some light into the higher UDS positivity in Black patients seen in our study. If we remove marijuana from the positive UDS results, the gap in UDS positivity between White and Black individuals may narrow (35% vs 36.5%, $p = 0.725$).

In our examination of ED utilization, Black patients made multiple visits for seizure-related issues, becoming more significant by the fourth visit. This aligned with the Fantaneau et al. study [6], which reported that Black patients had a high number of ED visits for seizure evaluation due to running out of or missing their antiseizure medications (ASMs). Likewise, Fantaneau et al. [6] found that these repeated visits were linked to limited access to regular care within the hospital system, prompting more frequent visits to the ED. Additionally, mistrust of physicians among Black patients may further contribute to this issue [4,16]. Improving outpatient follow-up access and rates could improve the relationship between patients and healthcare providers and, thus, reduce the number of ED visits for seizures.

In our cohort, ED disposition did not differ across racial and ethnic groups—except that Black patients, compared to White patients, had higher rates of NICU admissions. Only one Black patient was discharged against medical advice. National studies show that Black and Hispanic patients have greater odds of being discharged against medical advice compared to White patients; however, this analysis is not specific to UDS-only [17]. We found no racial difference in EEG testing rates. In contrast, Schiltz et al. [18] demonstrated lower EEG monitoring rates in Black and male participants in a medical center in California. These patients were primarily uninsured or publicly insured with Medicaid and Medicare compared to privately insured patients [18].

Regarding language, our cohort found English speakers more likely to undergo UDS testing and have higher UDS positivity rates. This also correlates with the higher number of non-Hispanic individuals testing positive for UDS. Notably, no UDS studies based on language were found in the literature so we cannot compare our own results.

Racial disparities in medical testing and treatment have significant implications for health outcomes, patient trust, and healthcare equity in the United States. Disparities in testing can lead to delayed treatment, missed diagnoses, undertreatment of life-threatening conditions, and poorer health outcomes for minority groups, including Black and Hispanic patients [4,19]. Repeated biases, over-testing for substance use, or under-testing for medical conditions can decrease trust in healthcare providers and institutions or misattribute symptoms to the positive UDS [16,19–21]. Distrust may prevent patients from seeking medical attention, leading to worse health outcomes and increased emergency utilization. Black and Hispanic patients are more likely to rely on emergency departments for primary care needs due to limited access to outpatient services [6]. Structural inequities, including limited access to primary care and lower socioeconomic status, restrict opportunities for preventive and continuous care [19]. To close the gap between racial and ethnic disparities in UDS for seizure patients in the ED, hospitals and institutions should establish standardized, evidence-based protocols, implement feedback systems, and address underlying social determinants of health. Adopting these steps will foster equity, enhance patients' trust, and ensure that testing is clinically justified rather than influenced by bias [22].

Our retrospective study had several limitations. First, there was a documentation bias due to incomplete or inaccurate data from existing records, particularly race documentation —could be self-reported or assigned by administrative staff—introducing variability and potential misclassification. Second is temporal and geographic limitations as this study was conducted at a single medical center and over a defined period; thus, the findings may not be generalizable to other institutions or populations. Third, due to the retrospective nature, the study is susceptible to unmeasured confounders and selection biases that may have influenced both the exposure and outcome, which we could not fully control due to limitations in the available data. There was also limited information on clinical rationale for ordering UDS as well as potential confounding factors such as socioeconomic status or clinical severity which were not discussed.

## Conclusion

Our study revealed that Black patients presenting to the ED with suspected seizures were more likely to undergo UDS compared to White patients. We also found that Black patients visited the ED more frequently to be evaluated for seizures. No ethnic disparities were found in our study. These findings highlight healthcare disparities disproportionately affecting racial minorities in a single-institution cohort. Additional research studies are necessary to address the gaps in seizure care.

## Author contributions

**Conceptualization:** Gloria Ortiz-Guerrero, Carol M. Ulloa.

**Data curation:** Gloria Ortiz-Guerrero, Maggie Logan, Utku Uysal.

**Formal analysis:** Gloria Ortiz-Guerrero, Utku Uysal.

**Investigation:** Gloria Ortiz-Guerrero, Utku Uysal, Carol M. Ulloa.

**Methodology:** Gloria Ortiz-Guerrero, Carol M. Ulloa.

**Writing – original draft:** Gloria Ortiz-Guerrero.

**Writing – review & editing:** Gloria Ortiz-Guerrero, Maggie Logan, Utku Uysal, Patrick Landazuri, Lakeshia Jackson, Carol M. Ulloa.

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
