## [Decision Letter · Decision Letter 0]

16 May 2025

Dear Dr. Ortiz Guerrero,

Thank you for submitting your manuscript to PLOS ONE. After careful consideration, we feel that it has merit but does not fully meet PLOS ONE’s publication criteria as it currently stands. Therefore, we invite you to submit a revised version of the manuscript that addresses the points raised during the review process.

We look forward to receiving your revised manuscript.

Kind regards,

David Wampler

Academic Editor

PLOS ONE

Journal Requirements:

3. Please upload a new copy of Figure 3 as the detail is not clear. Please follow the link for more information: https://blogs.plos.org/plos/2019/06/looking-good-tips-for-creating-your-plos-figures-graphics/ .

Additional Editor Comments:

Dear Dr. Guerrero

I would like to apologize for the delay in getting your reviewer comments. The current environment of recruiting high quality reviews has become challenging.

From the comments below, reviews are favorable and your paper has merit, and with a some significant work I would be happy to reconsider for publication in PLOS One.

I look for forward to your revision,

David Wampler, PhD

Editor

Reviewers' comments:

Reviewer's Responses to Questions

**Comments to the Author**

1. Is the manuscript technically sound, and do the data support the conclusions?

Reviewer #1: Yes

2. Has the statistical analysis been performed appropriately and rigorously?

Reviewer #1: Yes

3. Have the authors made all data underlying the findings in their manuscript fully available?

Reviewer #1: Yes

4. Is the manuscript presented in an intelligible fashion and written in standard English?

Reviewer #1: Yes

Reviewer #1: Reviewer comment and suggestions

Title

• The title is clear and informative, accurately reflecting the study's focus on racial disparities in urine drug screening (UDS) among patients presenting with suspected seizures.

Abstract:

• An abstract summarizing the background, methods, results, and conclusion will enhance accessibility and provide quick insight into the study’s contributions. An abstract following PLOS guidelines should also include keywords.

Introduction:

• The introduction could benefit from a more thorough literature review. Acknowledging existing studies on UDS disparities would strengthen the foundation for this research. Consider including the rationale for focusing specifically on UDS and the clinical implications of equitable access to this testing.

Methods:

• A clear description of sample selection criteria is provided, but details about exclusion criteria or specific demographic data (e.g., age ranges, comorbid conditions) would enhance reproducibility.

• Information regarding the statistical analysis might be more detailed. Indicating confidence intervals and effect sizes for the Chi-square results would provide a clearer picture of the clinical significance of the findings.

Results:

• The presentation of data is straightforward, with clear comparisons drawn between racial groups. However, including visual aids such as tables or graphs to present results (e.g., rates of UDS testing and positivity by race) could improve clarity.

• The narrative mentions UDS substance positivity but does not elaborate on the potential clinical implications of these findings, such as differences in management or treatment outcomes based on detected substances.

Discussion:

• The discussion provides a solid interpretation of findings but could incorporate broader implications of racial disparities in medical testing and treatment. Consider exploring the potential reasons for observed disparities, including systemic biases or differences in healthcare access.

• Addressing the limitations of the study, such as potential biases in retrospective data analysis or unmeasured confounders would enhance the transparency of the research.

• The conclusion reiterates the main findings but could benefit from a call to action, such as recommendations for addressing these disparities in clinical practice or suggestions for future research.

References:

• Ensure that all references are current and relevant, following PLOS guidelines for citation style. A comprehensive reference list will enhance the study's credibility.

General Formatting:

• Ensure that the manuscript follows PLOS ONE formatting guidelines, including specific font sizes, headings, and citation styles. This attention to detail is essential for submission readiness.

Ethical Considerations:

• Consider including a statement confirming institutional review board approval and patient consent where applicable.

In summary, this study addresses an important topic concerning racial disparities in UDS among patients with suspected seizures. Improvements in clarity and depth of analysis, alongside adherence to PLOS guidelines, will strengthen the manuscript's overall impact and quality.

**Do you want your identity to be public for this peer review?** For information about this choice, including consent withdrawal, please see our Privacy Policy

Reviewer #1: **Yes: ** REHEMA ABDALLAH

---

## [Author Response · Author response to Decision Letter 1]

21 Jun 2025

Journal Requirements:

1. Please ensure that your manuscript meets PLOS ONE's style requirements, including those for file naming. The manuscript has been updated to meet PLOS ONE’s style requirements.

2. We note that your Data Availability Statement is currently as follows: [All relevant data are within the manuscript]

I updated this option since all our relevant data is within the manuscript.

3. Please upload a new copy of Figure 3 as the detail is not clear. Please follow the link for more information: We have updated figure 3 which now corresponds to figure 4.

---

## [Decision Letter · Decision Letter 1]

15 Aug 2025

Dear Dr. Ortiz Guerrero,

We look forward to receiving your revised manuscript.

Kind regards,

David Wampler

Academic Editor

PLOS ONE

Journal Requirements:

Additional Editor Comments:

Dear Guerrero,

Congratulations, with only a few minor edits I am prepared to recommend your work for publication in PLOS One. Reviewers just ask that you tighten up a few disclosures.

Again congrats, Well done.

David Wampler, PhD

Academic Editor, PLOS One

Reviewers' comments:

Reviewer's Responses to Questions

**Comments to the Author**

Reviewer #1: All comments have been addressed

2. Is the manuscript technically sound, and do the data support the conclusions?

Reviewer #1: Yes

3. Has the statistical analysis been performed appropriately and rigorously?

Reviewer #1: I Don't Know

4. Have the authors made all data underlying the findings in their manuscript fully available?

Reviewer #1: Yes

5. Is the manuscript presented in an intelligible fashion and written in standard English?

Reviewer #1: Yes

Reviewer #1: Reviewer comment and suggestions

This article presents a retrospective analysis examining racial and ethnic disparities in urine drug screening (UDS) among patients with suspected seizures in the emergency department (ED).However some of issues need more clarifications.

Strengths of the study include:

• A sizable sample over multiple years.

• Focus on an understudied area of healthcare disparities.

• Use of objective data from electronic medical records.

Limitations include:

• Single-center design, which may limit generalizability.

• Retrospective nature, which cannot establish causality.

• Limited information on clinical rationale for ordering UDS.

• Potential confounding factors such as socioeconomic status or clinical severity were not discussed.

**Do you want your identity to be public for this peer review?** For information about this choice, including consent withdrawal, please see our Privacy Policy

Reviewer #1: **Yes: ** rehema abdallah

---

## [Author Response · Author response to Decision Letter 2]

19 Aug 2025

Thanks for considering our manuscript for publication. Below are the modifications we made to our manuscript as requested:

1. We added the financial disclosures to the cover letter as requested.

2. We expanded the study limitations to include the suggestion provided by Dr. Abdallah. These changes are highlighted in tracked changes on page 12, between line 239 and 240.

3. We used Preflight Analysis and Conversion Engine (PACE) to ensure figures meet PLOS requirements.

---

## [Editor Report · Decision Letter 2]

24 Aug 2025

Racial disparities in urine drug screening among seizure patients in the emergency department

PONE-D-24-55625R2

Dear Dr. Ortiz Guerrero,

We’re pleased to inform you that your manuscript has been judged scientifically suitable for publication and will be formally accepted for publication once it meets all outstanding technical requirements.

Kind regards,

David Wampler

Academic Editor

PLOS ONE
---

## [Editor Report · Acceptance letter]

PONE-D-24-55625R2

PLOS ONE

Dear Dr. Ortiz-Guerrero,

I'm pleased to inform you that your manuscript has been deemed suitable for publication in PLOS ONE. Congratulations! Your manuscript is now being handed over to our production team.

Kind regards,

on behalf of

Dr. David Wampler

Academic Editor

PLOS ONE